# *MIR390* Is Involved in Regulating Anthracnose Resistance in Apple

**DOI:** 10.3390/plants11233299

**Published:** 2022-11-29

**Authors:** Jiajun Shi, Qiu Jiang, Shuyuan Zhang, Xinyu Dai, Feng Wang, Yue Ma

**Affiliations:** 1College of Horticulture, Shenyang Agricultural University, Shenyang 110866, China; 2Liaoning Institute of Pomology, Xiongyue 115009, China; 3College of Plant Protection, Shenyang Agricultural University, Shenyang 110866, China

**Keywords:** apple, *MIR390*, disease resistance, anthracnose

## Abstract

As an important cash crop in China, apple has a good flavor and is rich in nutrients. Fungal attacks have become a major obstacle in apple cultivation. *Colletotrichum gloeosporioides* is one of the most devastating fungal pathogens in apple. Thus, discovering resistance genes in response to *C. gloeosporioides* may aid in designing safer control strategies and facilitate the development of apple resistance breeding. A previous study reported that ‘Hanfu’ autotetraploid apple displayed higher *C. gloeosporioides* resistance than ‘Hanfu’ apple, and the expression level of *mdm-MIR390b* was significantly upregulated in autotetraploid plants compared to that in ‘Hanfu’ plants, as demonstrated by digital gene expression (DGE) analysis. It is still unclear, however, whether *mdm-MIR390b* regulates apple anthracnose resistance. Apple *MIR390b* was transformed into apple ‘GL-3′ plants to identify the functions of *mdm-MIR390b* in anthracnose resistance. *C. gloeosporioides* treatment analysis indicated that the overexpression of *mdm-MIR390b* reduced fungal damage to apple leaves and fruit. Physiology analysis showed that *mdm-MIR390b* increased *C. gloeosporioides* resistance by improving superoxide dismutase (SOD) and peroxidase (POD) activity to alleviate the damage caused by O_2_^−^ and H_2_O_2_. Our results demonstrate that *mdm-MIR390b* can improve apple plants’ anthracnose resistance.

## 1. Introduction

Apple (*Malus × domestica*), a delicious fresh and processed fruit, is cultivated in many countries worldwide [1]. The spread of fungal diseases is one of the most important factors that restricts the development of apple production [2]. *Colletotrichum gloeosporioides* is a destructive fungus that causes early severe defoliation and severely weakens tree vigor. Attenuating the effects of *C. gloeosporioides* would be significant for apple production. The use of chemical fungicides is effective. However, it causes serious damage to the environment [3]. In order to reduce the pollution of the environment, improving the defensive ability of plants has become a good choice. Therefore, finding plant disease resistance genes for molecular breeding is an effective means to prevent fungal diseases.

MicroRNAs (miRNAs) are small, noncoding, endogenous RNA molecules that are 20-24 nucleotides (nt) in length [4,5]. They have been proven as new regulators, controlling post-transcriptional mRNA stability, providing epigenetic modifications to characteristic regions of the genome, or allowing translation by complementarily binding to target nucleic acids [4,6,7]. In plants, miRNA biogenesis and the modes of action have been specifically described. Most *MIR* genes are transcribed as independent units by DNA-dependent RNA Polymerase II (Pol II), while some miRNA precursors are produced from introns of the genes encoding proteins [8]. Primary transcripts of *MIRs* (pri-miRNAs) from typical *MIR* genes are modified similarly to mRNA and folded into hairpin structures [5]. Then, pri-miRNAs are recognized and processed by the dicing complex containing DICER-LIKE RNase III endonucleases (DCLs), the RNA-binding protein HYPONASTIC LEAVES 1 (HYL1), and the zinc-finger protein SERRATE (SE) to produce miRNA:miRNA* duplexes [9,10,11]. After the 2′-OH position in the duplex is methylated by the small RNA methyltransferase HUA Enhancer 1 (HEN1), the duplex is exported to the cytoplasm by HASTY (HST) [12]. The mature miRNA strand in the duplex binds to different ARGONAUTE (AGO) proteins to form the RNA-induced silencing complex (RISC) for targets, and the miRNA* strand is damaged by SMALL RNA DEGRADING NUCLEASE (SDN) [5]. Eventually, mature miRNAs downregulate or repress their targets at the post-transcriptional or translational level and are therefore involved in plant growth and development and stress responses.

Plant miR390, an ancient and highly conserved miRNA, is the only one present in many species and at high abundances [13]. Two miR390s (ath-miR390a and ath-miR390b) were first discovered in *Arabidopsis thaliana* in the Arabidopsis Small RNA Project (ASRP) [14]. Later, miR390s were found in a variety of plants. There was one osa-miR390 in rice [15], three nta-miR390s in tobacco [16], two sly-miR390s in tomato [17], and two zma-miR390s in maize [18]. Many studies have shown that miR390 acts on target genes in a different regulatory manner compared to other miRNAs. It can produce a 21-nt-long *trans*-acting small interfering RNA (tasiRNA) by acting on a *trans*-acting siRNA gene (*TAS*), and then targets auxin response factor (ARF) [19]. The transcript transcribed by Pol II from the *TAS3* gene is first cleaved by miR390 and the AGO1/7 complex to generate single-stranded RNA in a typical “two-hit” mode [20,21,22]. The single-stranded RNA is converted into double-stranded RNA by the plant-RNA-dependent RNA polymerase 6-centred (RDR6) complex [23,24]. Then, the double-stranded RNA bound to dsRNA-binding protein 4 (DRB4) is cleaved by DCL4 into short RNA fragments linked end-to-end, namely tasiRNA3 [19,25,26]. Finally, tasiRNA3 negatively regulates the expression of auxin response factor 2/3/4 (ARF2/3/4) [27,28,29]. In Arabidopsis, miR390 is specifically expressed at the sites of lateral root (LR) initiation to regulate the timing of LR growth through the miR390-tasiRNA3-*ARF2/3/4* pathway [26]. The miR390-tasiRNA-*ARF4* regulatory module contributes to auxin responses in LR developmental processes [30]. In flowering development, the miR390-tasiRNA3-*ARF4* pathway delays woodland strawberry florescence by targeting two genes related to flowering (*FveFUL* and *FveAP1*) [31]. In poplar, the miR390/*TAS3*/*ARFs* module acts as a key regulator of increased salt tolerance via modulating the auxin signaling [32]. In plants, miR390 plays a significant role in diverse processes, including root growth, floral development, and abiotic stress responses. However, research on the fungal responses of miR390 is still rare.

In our earlier study, ‘Hanfu’ apple trees displayed lower *C. gloeosporioides* resistance than autotetraploid apple trees [33]. DGE analysis showed that *MIR390b* was differentially expressed in ‘Hanfu’ and autotetraploid apple trees [34]. In this study, bioinformatics analysis of miR390 was performed and showed that miR390 sequences were highly conservative in plants. When ‘Hanfu’ plants were attacked by *C. gloeosporioides*, we found that *mdm-MIR390b* could be dramatically induced. Then, a transgenic method was used to investigate the role of *mdm-MIR390b* in *C. gloeosporioides* infection responses in apple. Overexpression of *mdm-MIR390b* significantly enhanced apple anthracnose resistance under the *C. gloeosporioides* stress condition compared with ‘GL-3′ control plants. Under *C. gloeosporioides* infection, overexpressing-*MIR390b* lines showed improved SOD and POD activity against pathogen-induced reactive oxygen species (ROS) accumulation. The expression levels of disease resistance genes (*PRs*) were upregulated in plants overexpressing *mdm-MIR390b*. These results provide new insights into the connection between *MIR390* and fungal stress in apple.

## 2. Results

### 2.1. Bioinformatics Analysis of miR390

In the miRbase database, six *mdm-MIR390s* were found and each *mdm-MIR390* can produce a mature miRNA (Table 1). Analysis of the phylogenetic tree found that *mdm-MIR390b*, *mdm-MIR390e*, and *mdm-MIR390f* showed strong homology, and *mdm-MIR390a*, *mdm-MIR390c*, and *mdm-MIR390d* showed a closer relationship. Six *mdm-MIR390s* were clustered in the same branch. They showed a more distant homology to *osa-MIR390*, *zma-MIR390s*, and *sbi-MIR390* (Figure 1a). Furthermore, *MIR390s* in dicot plants displayed higher sequence identity with *mdm-MIR390s* than those in monocot plants.

Stem-loop structures of six precursor sequences were predicted by minimum free energy using RNAfold. As shown in Figure 1b, mature mdm-miR390s (21 nt) were generated from the 5’ arm of the stem-loop structure and there is one mismatched base pair in the miRNA:miRNA*duplex. The end of the miRNA:miRNA* duplex has a 2 nt overhang structure, which resulted from Dicer cleavage [35]. The results revealed that *mdm-MIR390s* have universal features of pre-miRNAs.

In fifteen species, the predicted miR390s contained two typical sequences produced from the 5′ arm of the *MIR390* stem-loop. Most of them have the same sequence (5′-AAGCUCAGGAGGGAUAGCGCC-3′). Six mdm-miR390s also show this conserved sequence, and only three miR390s (nta-miR390a, sly-miR390a-5p, and stu-miR390) are different sequences (Figure 2a). The analysis of miR390 mature sequences’ alignment was visualized by WebLogo. This result showed that mature sequences had high identity, containing 20 bases that were completely conserved (Figure 2b).

### 2.2. Expression Patterns of mdm-MIR390b in Apple

Previously, we found that the ‘Hanfu’ autotetraploid has stronger anthracnose resistance than ‘Hanfu’ [33]. DGE analysis of ‘Hanfu’ and autotetraploid apple trees revealed that *mdm-MIR390b* has a higher expression level in autotetraploid apple plants [34]. Thus, in ‘Hanfu’ and autotetraploid apple leaves, quantitative reverse transcription polymerase chain reaction (qRT-PCR) analysis was performed to examine the expression of *mdm-MIR390b*. Compared with ‘Hanfu’ apple leaves, the expression level of *mdm-MIR390b* was statistically increased, with an over three-fold higher relative expression level in autotetraploid apple leaves (Figure 3a). Meanwhile, we observed the expression of *mdm-MIR390a-f* in the ‘Hanfu’ leaf and fruit. In the leaf, the relative expression level of *mdm-MIR390b* was higher than that of five *mdm-MIR390s*. In the fruit, *mdm-MIR390a* and *mdm-MIR390b* displayed higher expression levels (Appendix A). To examine fungus-dependent changes in the expression of *mdm-MIR390b*, qRT-PCR was used to analyze its expression after *C. gloeosporioides* treatment in ‘Hanfu’ apple plants. Apple *MIR390b* expression was upregulated after *C. gloeosporioides* infection and peaked with an approximately six-fold higher relative expression level at 2 d. The relative expression levels of mdm-miR390 and tasiRNA3 were also examined. They showed a similar trend as the expression level of *mdm-MIR390b* under *C. gloeosporioides* treatment (Figure 3b).

### 2.3. Overexpressing MIR390b in Apple ‘GL-3′

To investigate the function of *mdm-MIR390b*, we overexpressed *mdm-MIR390b* in apple ‘GL-3′ (Figure 4a). Three *MIR390b*-overexpressing lines under the control of the CaMV 35S promoter (*MIR390b*-line1, *MIR390b*-line2, and *MIR390b*-line3) were generated. The relative expression level of *mdm-MIR390b* was analyzed by qRT-PCR in apple ‘GL-3′ plants and *MIR390b*-overexpressing lines. As shown in Figure 4b, in the three *mdm-MIR390b* overexpression lines, the relative expression of *mdm-MIR390b* was significantly higher than that of ‘GL-3′. The expression level was, respectively, around 5.5, 4.0, and 5.0 times higher in *MIR390b*-line1, *MIR390b*-line2, and *MIR390b*-line3 than in ‘GL-3′. The expression level of mdm-miR390 in *MIR390b*-overexpressing plants was also significantly higher than that in ‘GL-3′ plants. Because the expression levels of *mdm-MIR390b* and mdm-miR390 in *MIR390b*-line1 and *MIR390b*-line3 plants were higher than those in *MIR390b*-line2 plants, the *MIR390b*-line1 and *MIR390b*-line3 plants were subjected to further investigation. Leaf regeneration experiments were performed using ‘GL-3′ and transgenic apple lines. On the right side of the glass dishes, the leaves of *mdm-MIR390b*-overexpressing lines again grew green resistant buds. However, on the left side of the dishes, the control leaves became white and lost their vitality (Figure 4c). These results indicated that the pRI101-*MIR390b* vector was transformed into apple ‘GL-3′ and three *mdm-MIR390b*-overexpressing lines were successfully obtained.

### 2.4. Overexpression of mdm-MIR390b Gene Elevates Apple Anthracnose Resistance

To clarify the function of *mdm-MIR390b* in *C. gloeosporioides* resistance, we examined the growth changes in *mdm-MIR390b*-overexpressing and ‘GL-3′ apple plants under *C. gloeosporioides*-stressed conditions. After apple plants were sprayed with fungal suspension, discrete small lesions were observed between *mdm-MIR390b*-overexpressing and ‘GL-3′ apple plants on the third day. Apple *MIR390b*-line1 and *MIR390b*-line3 plants displayed weaker disease situations than ‘GL-3′ plants. On the fifth day, larger brown spots were found in the leaves of ‘GL-3′, *MIR390b*-line1, and *MIR390b*-line3 plants. The control plants showed more severe disease conditions than overexpressing transgenic apple plants (Figure 5a). Quantitative analysis confirmed that the disease indexes of *MIR390b*-overexpressing plants were significantly lower than those of ‘GL-3′ plants after *C. gloeosporioides* treatment (Figure 5b). In apple fruit, the lesion diameter of the control fruit was greater than that of *MIR390b*-overexpressing plants. The overexpression of *mdm-MIR390b* inhibited the brown lesions that developed rapidly and the abundant conidia that were produced around the injection site. In contrast, the control apples exhibited more severe disease symptoms (Figure 5c,d). When the fruit were cut through the centers of lesions, V-shaped necrotic tissue areas were also observed. The lesions of control apples were 1.68-fold deeper than those in *mdm-MIR390b*-overexpressing apples (Figure 5e,f). These results indicate that the overexpression of *mdm-MIR390b* can enhance the anthracnose resistance of apple leaves and fruit.

### 2.5. Physiological Changes in Apple Leaves after C. gloeosporioides Infection

ROS have important roles in pathogen–plant interaction and are involved in hypersensitive responses and signal transduction. To investigate the actions of leaves against *C. gloeosporioides* infection, O_2_^−^ was detected in ‘GL-3′ and *MIR390b*-overexpressing plants by a chemical tissue staining method. Via nitro blue tetrazolium chloride (NBT) staining, more blue precipitations in ‘GL-3′ leaves were observed, indicating a greater level of O_2_^−^ accumulation than in *MIR390b*-overexpressing leaves exposed to *C. gloeosporioides* stress (Figure 6a). SOD catalyzes the conversion of O_2_^−^ into H_2_O_2_ and POD uses H_2_O_2_ as an electron acceptor to directly oxidize phenols or amines. They are the first line of defense against oxidative damage and remove excess free radicals in plants and improve the stress resistance of plants. Accordingly, SOD and POD activity was measured in ‘GL-3′ and *MIR390b*-overexpressing plants. Under normal conditions, there was no significant difference in SOD and POD activity between ‘GL-3′ and *MIR390b*-overexpressing plants. Under *C. gloeosporioides* treatment, the antioxidant enzyme activity of all apple plants increased, and compared with ‘GL-3′, the SOD and POD activity of *MIR390b*-line1 and *MIR390b*-line3 plants increased at 3 d (Figure 6b,c). In agreement with the results of NBT staining, the O_2_^−^ content was much higher in the control plants than in *MIR390b*-overexpressing plants under *C. gloeosporioides* treatment. These results indicated that *MIR390b*-overexpressing plants probably achieved enhanced *C. gloeosporioides* resistance by increasing the activity of antioxidant enzymes in vivo in response to *C. gloeosporioides* stress.

### 2.6. Changes in Disease-Resistant Genes in MIR390b-Overexpressing Plants

Pathogenesis-related (PR) genes play unique roles in pathogen defense, and their expression is associated with enhanced resistance to pathogens [36,37]. To obtain clues regarding the molecular mechanism of *mdm-MIR390b*-mediated anthracnose resistance, the expression of four previously known *C. gloeosporioides* response genes in *MIR390b*-line1 and *MIR390b*-line3 plants was examined by qRT-PCR analysis. The qRT-PCR results demonstrated that the transcription levels of the selected four marker genes (*PR2*, *PR3-1*, *PR10-1*, and *PR10-2*) were remarkably upregulated in *MIR390b*-overexpressing plants compared with the controls (Figure 7).

## 3. Discussion

Plant miRNAs ubiquitously exist in various organisms and play imperative roles in multiple life processes [38,39]. Defining the role of miRNAs in the plant stress response is a hotspot of research. Recently, the mechanism of miR156 regulating salt tolerance in apple has been clearly studied [40]. In wheat, tae-miR171a was validated to have an obvious response to drought memory by regulating the expression of target genes [41]. The molecular mechanism of miR482b expression in tomato infected by *Botrytis cinerea* was revealed [42]. In plants, miR390 is an ancient family of miRNAs and is conserved among species [13]. Our analysis showed that mature miR390 sequences were conserved and six mdm-miR390 sequences were consistent (Figure 2). In terms of LR growth, miR390 promotes LR growth and the cleavage of *TAS3* by AGO7 results in the production of tasiRNAs, which target mRNAs encoding ARFs [29,32]. In *Physcomitrella patens*, the overexpression of miR390 elevated miR390-triggered tasiRNA accumulation, decreased the level of tasiRNA targets, and caused the slower formation of gametophores [43]. It was reported that the overexpression of cgo-miR390s affected the development of reproductive organs in *Cymbidium goeringii* [44]. The effects of miR390 in plant growth and development have been investigated relatively clearly. However, studies on *MIR390* in the biotic stress response remain largely lacking. In addition, apple miRNAs’ regulatory networks were analyzed by deep small RNA-seq and the emergence, evolution, and diversification of miR390 in land plants were elaborated by a series of bioinformatics analyses [19,27]. Thus, the functions of miR390 should be further verified by the transgenic strategy in apple. ‘GL-3′ apple plants that overexpressed *mdm-MIR390b* were used to analyze the specific role of *mdm-MIR390b* in response to *C. gloeosporioides* infection (Figure 4). Our research clearly demonstrated that the overexpression of *mdm-MIR390b* in apple improved its resistance to *C. gloeosporioides* infection (Figure 5). In a recent study, researchers found that phased small RNA involved systemic signaling in plants. Two tasiRNA3 derived from *TAS3a* and synthesized within several hours of *Pseudomonas syringae* DC 3000 infection were the early mobile signal; they then cleaved *ARF2/3/4* to induce systemic acquired resistance (SAR) [45]. In *MIR390b*-overexpressing plants, the expression level of tasiRNA3 was higher compared with ‘GL-3′ plants. By contrast, qRT-PCR analysis of *MdARF2/3/4* showed that they had a lower expression level compared with ‘GL-3′ plants (Appendix A). We speculate that apple anthracnose resistance was elevated by the miR390-tasiRNA3-*ARFs* pathway. The role of this pathway during *C. gloeosporioides* infection needs to be elucidated.

ROS, as important signaling molecules, have been studied for their roles in plant immunity [46]. A transient increase in ROS induces the expression of resistance proteins in plants, but the continuous high accumulation of ROS can lead to negative effects in plant cells [47]. ROS were observed to accumulate in many plants after the perception of pathogens and are considered to be closely related to the defense response of the plant [48,49,50]. One approach to evaluating the abilities of plant defense is to observe the accumulation of ROS in plants. NBT staining is convenient and efficient, used in many studies used to detect ROS. After *C. gloeosporioides* treatment, NBT staining showed the higher accumulation of O_2_^−^ in ‘GL-3′ plants with respect to *MIR390b*-overexpressing plants (Figure 6a). Thus, the low level of ROS in the *MIR390b*-overexpressing plants probably enhanced the apple anthracnose resistance. A chloroplast-derived ROS burst destroyed the antioxidant enzyme system and caused cell death and leaf necrosis [51]. In research on postharvest longan, *Lasiodiplodia theobromae*-induced pericarp browning and disease development was due to the reduction of ROS scavenging ability and an increase in ROS production [52]. Excess ROS caused by pathogens severely harms plants, and plants have evolved using multiple methods to scavenge ROS [48,53]. One of the adaptive mechanisms is that the plants reduce the ROS content in vivo through antioxidant enzymes [48]. High SOD and POD activity is conducive to maintaining the physiological redox homeostasis to protect the plants from ROS-mediated damage. In the immune species *Chrysanthemum makinoi* var. *wakasaense*, SOD and POD activity increased significantly with *Puccnia horiana* inoculation [54]. The activity of SOD and POD increased at the sugarcane jointing stage, which was beneficial to deal with excessive ROS. The removal of superfluous ROS in sugarcane positively regulated resistance to smut [55]. High activity of ROS-scavenging enzymes (SOD) and high content of ascorbic acid and glutathione can inhibit the growth of *Peronophythora litchii* in harvested litchis, thereby enhancing disease resistance [56]. In agreement with these results, SOD and POD activity in *MIR390b*-overexpressing plants was higher than that in ‘GL-3′ plants (Figure 6b,c). In addition, the *MIR390b*-overexpressing fruit showed a small lesion diameter and depth, while the control fruit displayed serious necrotic tissue zones (Figure 5e,f). Hence, we speculated that the fruit might improve the anthracnose resistance by boosting SOD and POD activity. Active oxygen metabolism effectively prevented pathogens from stimulating membrane lipid peroxidation, causing the loss of cellular compartmentalization and the inhibition of photosynthetic electron transport [51,52]. *PR* genes were induced by pathogen infection and PR proteins may act as antimicrobial components in the plant defense response [36,57]. We found a remarkable upregulation of *PR* genes in *MIR390b*-overexpressing plants compared with ‘GL-3′ plants (Figure 7). The finding indicated that *mdm-MIR390b* strengthened apple’s resistance to *C. gloeosporioides* by improving the expression level of the *PR* gene. Thus, it is reasonable to conclude that *mdm-MIR390b* enhances resistance against pathogens by changing the activity of antioxidant enzymes and the expression levels of *PR* genes.

## 4. Materials and Methods

### 4.1. Plant Materials and Growth Conditions

Apple tissue culture plants including ‘GL-3′, ‘Hanfu’, and a ‘Hanfu’ autotetraploid were cultured in Murashige and Skoog culture medium supplemented with 0.1 mg/L gibberellic acid 3, 0.2 mg/L 3-indoleacetic acid, and 0.3 mg/L 6-benzylaminopurine (apple subculture medium). They were placed at (24 ± 1) °C under long-day conditions (16 h:8 h, light:dark). ‘Hanfu’ apple fruit were collected at a commercially mature stage from an orchard at Shenyang Agriculture University. ‘Hanfu’ was used in qRT-PCR and fungal inoculation experiments. The ‘Hanfu’ autotetraploid was used in qRT-PCR. ‘GL-3′ was used for *Agrobacterium*-mediated leaf transformation, histochemical staining, determination of antioxidant enzymes, and fungal inoculation experiments. The fruit without physical injuries and with uniform shape and size were selected for fungal inoculation experiments.

### 4.2. Pathogen Culture and Inoculation Method

The phytopathogenic fungus *Colletotrichum gloeosporioides* was cultured for 10 days on potato dextrose agar medium at 28 °C in the dark. Conidia were then collected in sterile water, and the suspension population was adjusted to 10^7^ spores/mL using a bacterial counting chamber. For fungal inoculation treatments, the plants of ‘Hanfu’ and ‘GL-3′ were sprayed with the above suspension. The inoculated plants were subsequently placed in the dark box at 25 °C for disease development. After 1, 2, 3, 4, and 5 d, plant leaves were used for photography or further experiments. At least three plants were used per treatment. The disease index was calculated according to a previous report [58].

### 4.3. Phylogenetic and Sequence Analysis

The precursor and mature sequences of the *MIR390* family were downloaded from a public miRNA database, miRbase (https://www.mirbase.org/, accessed on 20 November 2022) [59]. Apple miR390* sequences were obtained from a comprehensive functional plant miRNA database, PmiREN (https://www.pmiren.com/, accessed on 20 November 2022) [60]. *MIR390* precursors were used for phylogenetic analysis. The phylogenetic tree was constructed using MEGA X software with the Maximum Likelihood (ML) method under the Kimura 2-parameter model [61]. The primary tree was optimized by a web server, Evolview v3 [62].

Secondary structure prediction was performed with the web server RNAfold (http://rna.tbi.univie.ac.at/, accessed on 20 November 2022). Fold algorithms and basic options were set as follows: avoid isolated base pairs and minimum free energy (MFE) and partition function. Multiple sequence alignment of mature miR390 was executed by MEGA X and the result was processed using GeneDoc software. In fifteen plants, miR390 sequences were shown using the WebLogo online tool (http://weblogo.berkeley.edu/logo.cgi, accessed on 20 November 2022). The parameters were set to default.

ID numbers of apple *MIR390* homologous sequences were as follows: ath-*MIR390a* (MI0001000), ath-*MIR390b* (MI0001001), osa-*MIR390* (MI0001690), bna-*MIR390a* (MI0006447), bna-*MIR390b* (MI0006448), bna-*MIR390c* (MI0006449), vvi-*MIR390* (MI0006552), ghr-*MIR390a* (MI0005647), ghr-*MIR390b* (MI0005648), ghr-*MIR390c* (MI0005649), mtr-*MIR390* (MI0005586), sbi-*MIR390* (MI0010887), zma-*MIR390a* (MI0013209), zma-*MIR390b* (MI0013244), ppe-*MIR390* (MI0026099), nta-*MIR390a* (MI0021391), nta-*MIR390b* (MI0021392), nta-*MIR390c* (MI0021393), cme-*MIR390a* (MI0023238), cme-*MIR390b* (MI0018164), cme-*MIR390c* (MI0023239), sly-*MIR390a* (MI0029117), sly-*MIR390b* (MI0029123), stu-*MIR390* (MI0025988), fve-*MIR390a* (MI0036404), fve-*MIR390b* (MI0036405).

### 4.4. Nucleic Acid Extraction and qRT-PCR Analysis

Plants frozen in liquid nitrogen were grounded into powder and we extracted DNA and RNA using the cetyltrimethylammonium bromide (CTAB) extraction method [58]. The nucleic acid concentration, nucleic acid quality, and cDNA synthesis were performed according to established protocols [58].

qRT-PCR was performed on ABI QuantStudio™ 6 Flex real-time PCR instrument. Each reaction solution was a total volume of 10 μL containing 5μL 2 × UltraSYBR Mixture (CWBIO, CW2601M), 0.5 μL of cDNA, 0.5 μL of 0.2 μM *mdm-MIR390b*-QF and 0.2 μM *mdm-MIR390b*-QR primers (Appendix A), and 3.5 μL ddH_2_O. The reactions were initially polymerase activation at 95 °C for 5 min, followed by 40 cycles at 95 °C for 15s and 60 °C for 1 min. *MdEF-1α* (NCBI-ID: DQ341381) was an internal reference gene. For detecting the relative expression of mdm-miR390 and tasiRNA3, SuperReal PreMix (TIANGEN BIOTECH, FP206-02) was used. The previously reported methods were performed [31]. Apple 5SrRNA was selected as an internal reference. There were three biological and technical replicates. Relative expression levels of all examined genes and small RNA were calculated using the 2^−ΔΔCt^ method [63].

### 4.5. Vector Construction and Transformation into Apple

*mdm-MIR390b* (103 bp) was amplified from the genomic DNA of ‘Hanfu’ with the *mdm-MIR390b*-F and *mdm-MIR390b*-R primers listed in Appendix A by PCR. The product was ligated into a pRI101-AN binary plant transformation vector at double restriction sites (*Xba* Ⅰ and *Sac* Ⅰ) to construct pRI101-*MIR390b*. The PCR mixture and cycle conditions were obtained from the instruction manual (KOD, TOYOBO, Osaka, Japan).

pRI101-*MIR390b* was transformed into apple ‘GL-3′ by the *Agrobacterium tumefaciens*-mediated (EHA105) method [64]. Resistant buds from separate apple leaf explants were chosen as independent transgenic events for further identification.

### 4.6. Confirmation of Transgenic Plants

Healthy ‘GL-3′ and putative transgenic plants were selected. Their leaves were cut into two parts along the direction of the vertical vein, and we then placed leaf fragments on apple subculture medium supplemented with 25 mg/L kanamycin, 250 mg/L cefotaxim sodium salt, and 250 mg/L timentin. After 15 days in the dark, they were moved to a light culture.

### 4.7. Agroinfiltration in Apple Fruit

*A. tumefaciens* (EHA105) containing pRI101-AN and pRI101-*MIR390b* were used for the infection of fruit. The preparation of the infection suspension was performed as previously described [65]. The fruit were wounded approximately 5 mm deep with a 1 mL syringe, and then 200 μL *A. tumefaciens* suspension was injected into apple fruit using a needleless syringe. The wounded fruit were treated in darkness at 25 °C. After 5 d, each injected site was treated with a 6-mm-diameter mycelial disc of *C. gloeosporioides*. Next, all apple fruit were placed in damp plastic bags and stored at 25 °C in the dark. After 5 d of incubation, the lesion diameter and depth were measured. Ten fruit were infiltrated with each construct and five different fruit were evaluated for apple disease conditions.

### 4.8. Histochemical Staining

Four-week-old healthy apple plants were subjected to the inoculum suspension for 3 d. Then, the leaves of three independent plants were stained by NBT. The previously reported methods were performed [66,67]. The experiments were conducted three times.

### 4.9. Determination of Antioxidant Enzymes

The apple leaves were isolated from three independent apple plants treated with pathogen *C. gloeosporioides* for 3 d. SOD and POD activity was measured according to previously reported methods [68]. The experiments were conducted three times.

### 4.10. Statistical Analysis

The analysis of differences between the data was performed with the IBM SPSS 26 software (Chicago, IL, USA). All values are presented as mean ± SD (Standard Deviation) as indicated. The single asterisk (Student’s *t*-test) and different letters (one-way ANOVA Duncan’s test) indicate significant differences at the *p* < 0.05 level.

## 5. Conclusions

In conclusion, this study demonstrated that *mdm-MIR390b* plays critical roles during *C. gloeosporioides* infection in apple. The apple *MIR390b* gene showed inductive expression during *C. gloeosporioides* infection. *mdm-MIR390b*-overexpressing plants were created and had higher anthracnose resistance by ROS deoxidation and higher-level expression of *PR* than control plants. Moreover, *mdm-MIR390b* provides a reference for resistance breeding in apple.

## Figures and Tables

**Figure 1 plants-11-03299-f001:**
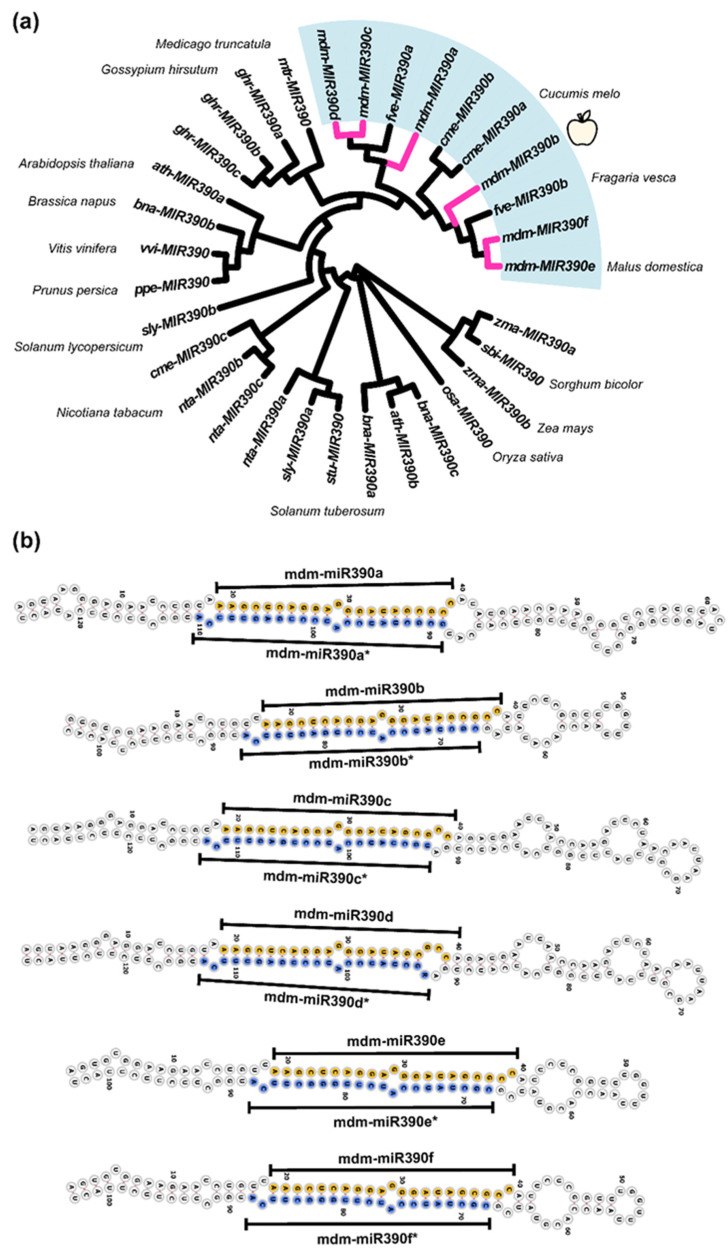
Identification and sequence analysis of *MIR390* family. (**a**) The phylogenetic analysis of *MIR390s* in plants. Bootstrap values were derived from 1000 replicates. The pink branches represent *mdm-MIR390s*. The cluster with the light blue background contains all *mdm-MIR390s.* The cosmic latte apple represents *mdm-MIR390b*. (**b**) *MIR390s* secondary structure analysis in apple.

**Figure 2 plants-11-03299-f002:**
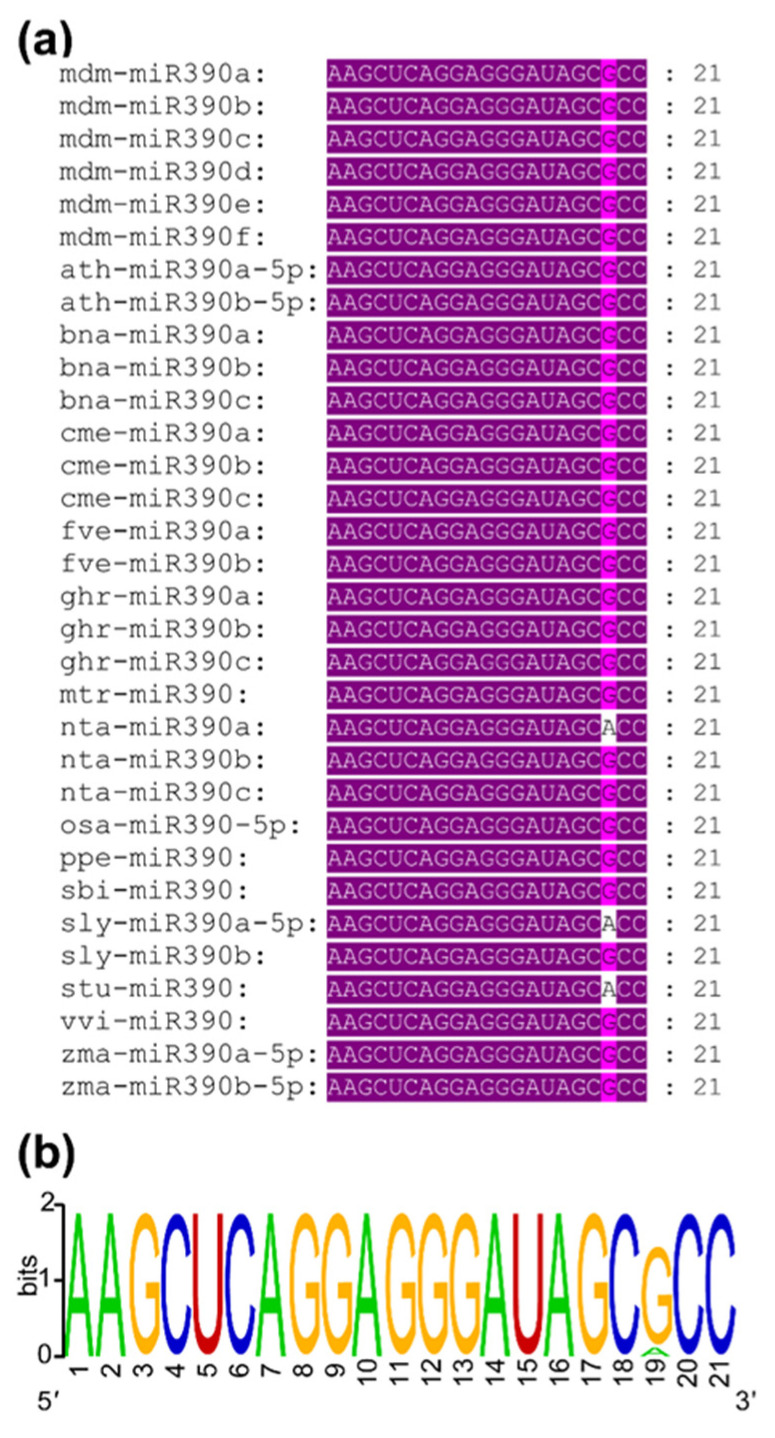
Mature sequence analysis of miR390s. (**a**) Sequence alignment of mdm-miR390s with other homologous sequences. (**b**) Conserved bases of miR390 mature sequences in fifteen species.

**Figure 3 plants-11-03299-f003:**
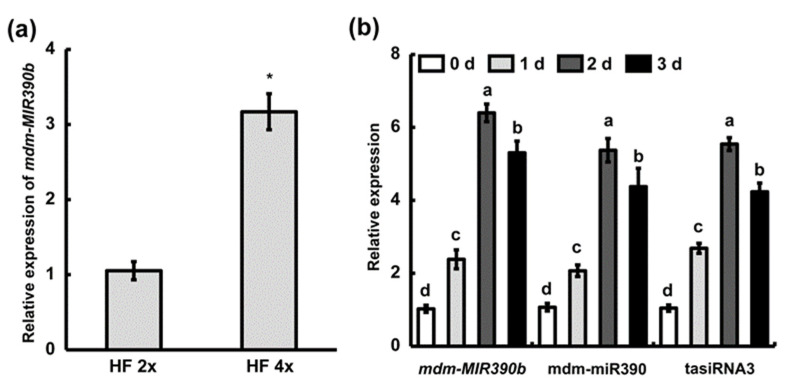
The analysis of the *mdm-MIR390b* gene, mdm-miR390, and tasiRNA3by qRT-PCR. (**a**) *mdm-MIR390b* expression was examined in ‘Hanfu’ and autotetraploid apple. HF 2x represents ‘Hanfu’. HF 4x represents ‘Hanfu’ autotetraploid. The vertical bars represent SDs (*n* = 3). ‘*’ represents *p* < 0.05 (Student’s *t*-test). (**b**) Expression level of *mdm-MIR390b*, mdm-miR390, and tasiRNA3 under *C. gloeosporioides* infection in ‘Hanfu’. The vertical bars represent SDs (*n* = 3). The different letters represent significant differences at *p* < 0.05 level.

**Figure 4 plants-11-03299-f004:**
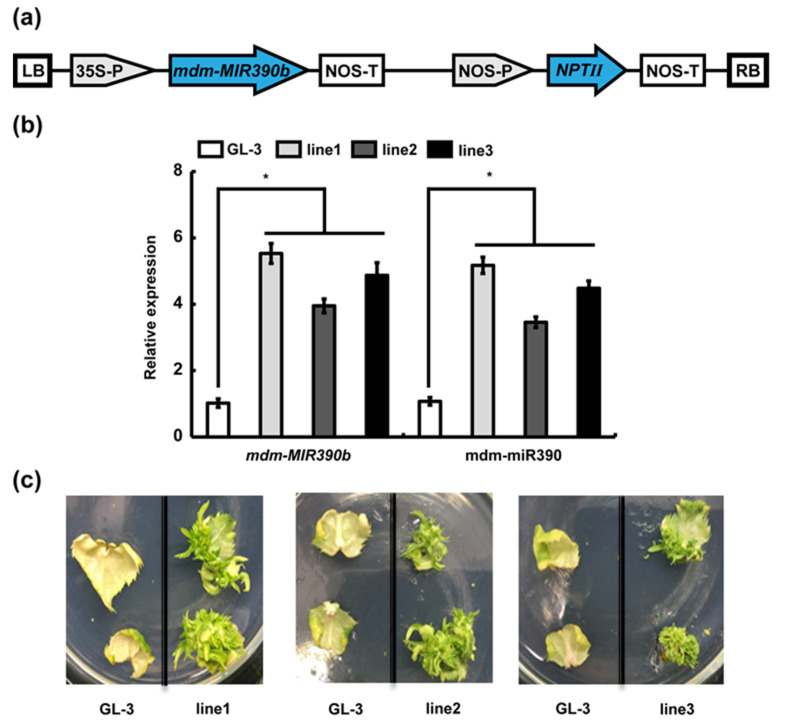
Identification and analysis of transgenic plants. (**a**) Structural diagram of pRI101-*MIR390b*. (**b**) Expression analysis of *mdm-MIR390b* gene and mdm-miR390 in ‘GL-3′ and *MIR390b*-overexpressing plants. The vertical bars represent SDs (*n* = 3). ‘*’ represents *p* < 0.05 (Student’s *t*-test). (**c**) Leaf regeneration observation of ‘GL-3′ and *MIR390b*-overexpressing plants.

**Figure 5 plants-11-03299-f005:**
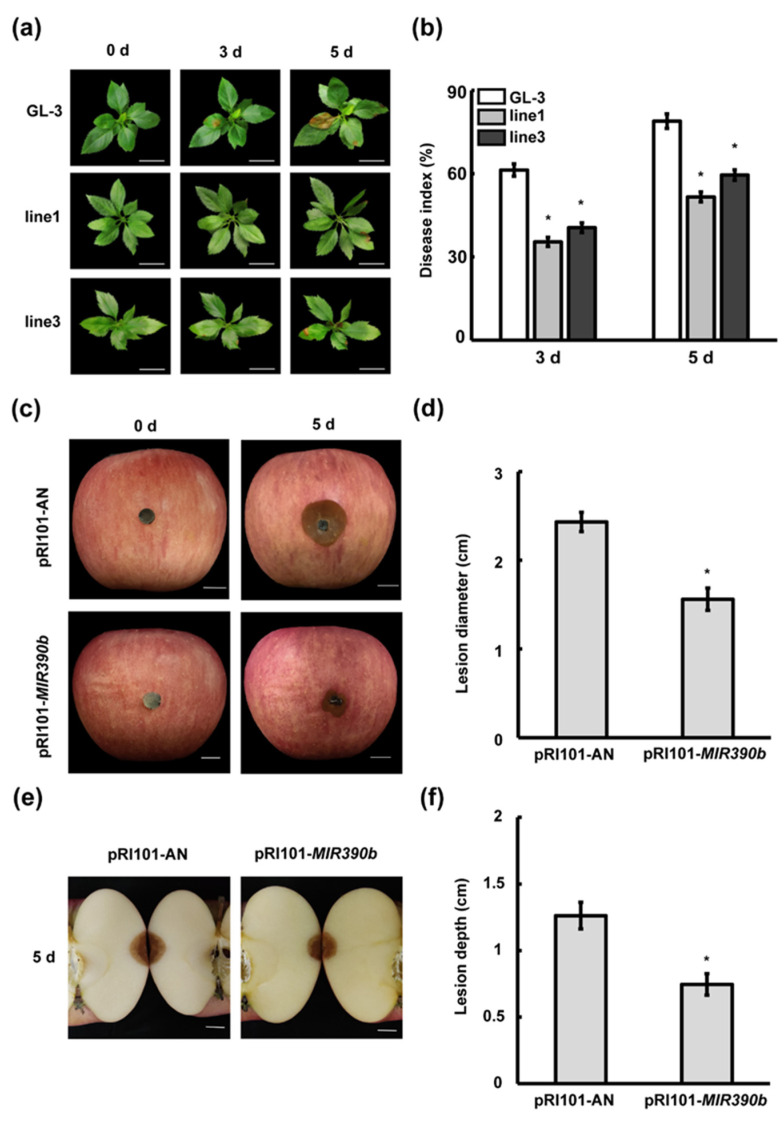
Apple *MIR390b* enhances resistance to *C. gloeosporioides* in leaves and fruit. (**a**) Leaf phenotype of ‘GL-3′ and *MIR390b*-overexpressing plants under *C. gloeosporioides* infection, white bars = 2 cm. (**b**) Disease index of ‘GL-3′ and *MIR390b*-overexpressing plants under *C. gloeosporioides* infection. The vertical bars represent SDs (*n* = 3). ‘*’ represents *p* < 0.05 (Student’s *t*-test). (**c**) Phenotype of apple fruit transiently expressing pRI101-*MIR390b* and empty vector (control) under *C. gloeosporioides* infection, white bars = 1 cm. (**d**) Lesion diameter of apple fruit after infection with *C. gloeosporioides* for 5 d. The vertical bars represent SDs (*n* = 3). ‘*’ represents *p* < 0.05 (Student’s *t*-test). (**e**) V-shaped necrotic tissue caused by inoculation of *C. gloeosporioides* in wound, white bars = 1 cm. (**f**) Fruit lesion depth caused by wound inoculation of *C. gloeosporioides*. The vertical bars represent SDs (*n* = 3). ‘*’ represents *p* < 0.05 (Student’s *t*-test).

**Figure 6 plants-11-03299-f006:**
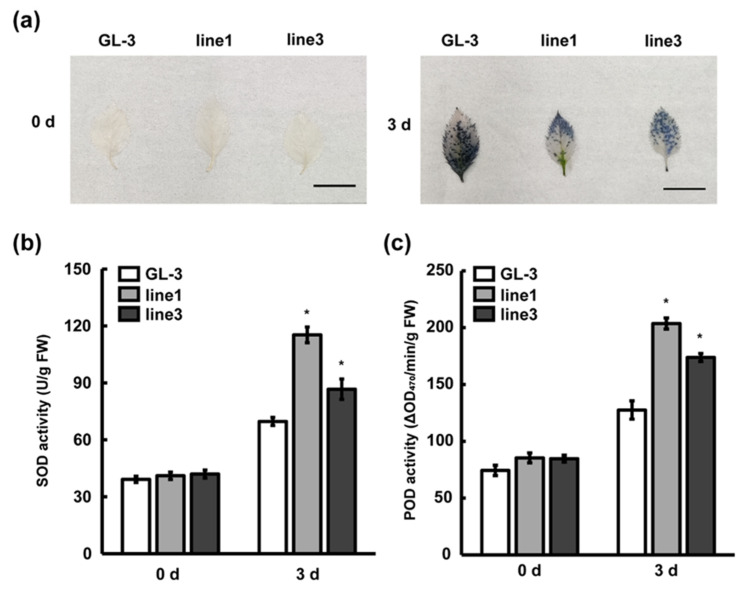
Changes in ROS content and biochemical characteristics during *C. gloeosporioides* infection. (**a**) NBT staining of ‘GL-3′ and *MIR390b*-overexpressing plants’ leaves under *C. gloeosporioides* infection, black bars = 1 cm. (**b**) The activity of SOD in ‘GL-3′ and *MIR390b*-overexpressing plants during *C. gloeosporioides* infection. The vertical bars represent SDs (*n* = 3). ‘*’ represents *p* < 0.05 (Student’s *t*-test). (**c**) The activity of POD in ‘GL-3′ and *MIR390b*-overexpressing plants during *C. gloeosporioides* infection. The vertical bars represent SDs (*n* = 3). ‘*’ represents *p* < 0.05 (Student’s *t*-test).

**Figure 7 plants-11-03299-f007:**
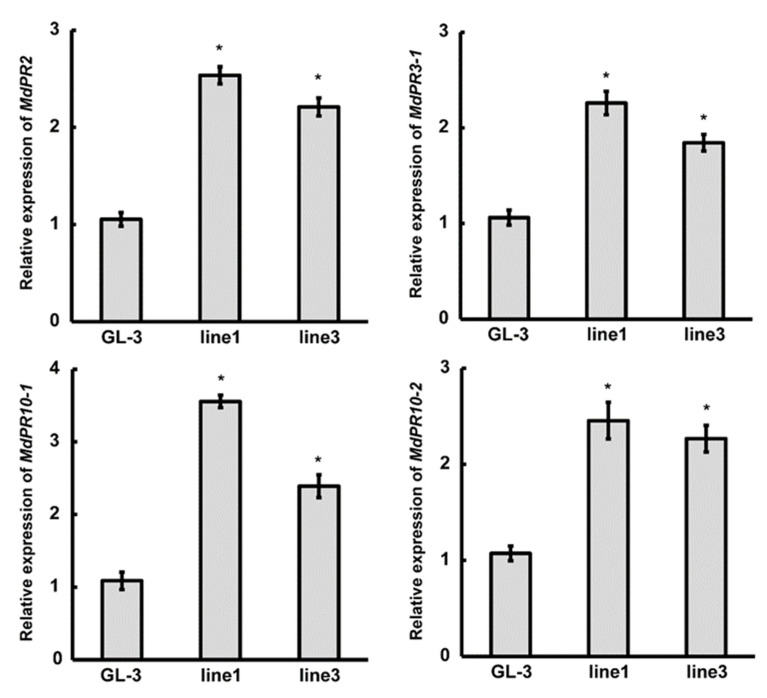
Expression analysis of disease-resistant marker genes in ‘GL-3′ and *MIR390b*-overexpressing plants. Leaves of 4-week-old ‘GL-3′, *MIR390b*-line1, and *MIR390b*-line3 plants were harvested. Expression of *PR2*, *PR3-1*, *PR10-1*, and *PR10-2* was measured in control and transgenic apple plants by qRT-PCR. The vertical bars represent SDs (*n* = 3). ‘*’ represents *p* < 0.05 (Student’s *t*-test).

**Table 1 plants-11-03299-t001:** Apple MIR390 and miR390 information.

Type	miRBase Name	miRBase ID	PmiREN Name
Precursors	*mdm-MIR390a*	MI0023073	*Mdo-MIR390e*
	*mdm-MIR390b*	MI0023074	*Mdo-MIR390d*
	*mdm-MIR390c*	MI0023075	*Mdo-MIR390c*
	*mdm-MIR390d*	MI0023076	*Mdo-MIR390f*
	*mdm-MIR390e*	MI0023077	*Mdo-MIR390a*
	*mdm-MIR390f*	MI0023078	*Mdo-MIR390b*
Mature miRNAs	mdm-miR390a	MIMAT0025969	Mdo-miR390e
	mdm-miR390b	MIMAT0025970	Mdo-miR390d
	mdm-miR390c	MIMAT0025971	Mdo-miR390c
	mdm-miR390d	MIMAT0025972	Mdo-miR390f
	mdm-miR390e	MIMAT0025973	Mdo-miR390a
	mdm-miR390f	MIMAT0025974	Mdo-miR390b

## Data Availability

Not applicable.

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
