# Peer review of "MIR390 Is Involved in Regulating Anthracnose Resistance in Apple"

_plants, 2022, doi:10.3390/plants11233299_

Round 1

Reviewer 1 Report

In this manuscript from Shi et al., the authors analyzed the bioinformation and expression pattern of miRNA390, showing that there are six conserved miRNA390 members in apple. miRNA390b was highly expressed in autotetraploid apple leaves and can be induced to express by C. gloeosporioides infection. The authors found that overexpression of miR390b in apple “GL-3” could enhance apple anthracnose resistance. In the overexpressing-390 transgenic apple lines, the activities of SOD and POD were increased, and the expression of disease-resistant genes were upregulated under C. gloeosporioides treatment. All these discoveries suggested that miR390b was involved in apple anthracnose resistance.

Shortness:  I think some additional experiments are necessary to improve the quality of this study.  This manuscript was not well written. Some sentences need to re-organization. I suggest ask a native speaker to edit the manuscript to improve it for publication.  The following are my major and minor comments:

Major comments:

1.       I want to ask the authors whether they could find out some targets of miRNA390 candidates by bioinformatics analysis, especially the candidates that maybe involved in pathogen resistance, and test their expression levels in miRNA390b-overexpressing transgenic lines?

2.       In the section “2.2. Expression patterns of mdm-MIR390b in apple”, “mdm-MIR390b has higher expression level in autotetraploid apple plants” has been reported in the authors’ previously paper. I don’t think fig3A is necessary here. I suggest that the authors analysis the six members of miRNA390 expression patterns in different organs, especially in leaves and fruits, which would greatly improve the novelty and originality of the study.

Minor comments:

1.       Line 28, delete “around the globe”.

2.       Line 29, to restrict the apple production.

3.       Line 34, delete “themselves”.

4.       Line 60, delete “and there were”.

5.       Line 97-98, the font format of “can produce a mature miRNA (Table 1). Analysis of phylogenetic tree found that” should be normal.

6.       Line 106, replace “position” with “base pair”.

7.       Line 121-122, replace “The light blue background contains all mdm-MIR390s” with “the cluster with light blue background contains all mdm-MIR390s”.

8.       Line 127, species

9.       Line 191, leaves phenotype or vegetative phenotype.

10.   Line 250, replace “as having” with “to have”.

11.   Line 253, conserved

12.   Line 255, what’s the “LR”?

13.   Line 277-278, this sentence is difficult to understand. Please re-organize it.

14.   Line 281, evaluate

Reviewer 2 Report

In this manuscript, the authors analyzed the sequences of the mdm-MIR390 family and investigated their role in apple anthracnose resistance. They found a higher MIR390b expression level in autotetraploid HanFu leaves, suggesting that it may play a role in anthracnose resistance. They further overexpressed this gene and found that overexpression of mdm-MIR390 enhanced resistance to C. gloeosporides, likely due to increased SOD activity and the upregulation of disease-resistant genes, such as MdPR2 and MdPR3-1. Overall, the data presented in the manuscript is of high quality and largely supports the statements.  For improvements of this manuscript, I have the following suggestions/comments:

1. In the whole paper, the authors only showed the expression level of mdm-MIR390. But the phenotypes they observed are missing mechanistic studies, especially related to the function of mdm-MIR390. For example, how about the mature miR390 levels? This is essential since the mature miRNAs are the functional elements. The authors need to perform a Northern blot or qRT-PCR to check the mature miRNA levels.

2. Following that, how about the expression level of TAS in all these conditions? 

Round 2

Reviewer 1 Report

I only have one minor comment: Line 292, "evaluating should be "evaluate".

Reviewer 2 Report

In this revised manuscript, the authors have addressed the questions raised by this reviewer, now it is suitable for publication.